# Guided Reinforcement Learning for Robust Multi-Contact Loco-Manipulation

**Jean-Pierre Sleiman**\*
ETH Zurich
jsleiman@ethz.ch

**Mayank Mittal**\*
ETH Zurich and NVIDIA
mittalma@ethz.ch

**Marco Hutter**
ETH Zurich
mahutter@ethz.ch

**Abstract:** Reinforcement learning (RL) often necessitates a meticulous Markov Decision Process (MDP) design tailored to each task. This work aims to address this challenge by proposing a systematic approach to behavior synthesis and control for multi-contact loco-manipulation tasks, such as navigating spring-loaded doors and manipulating heavy dishwashers. We define a task-independent MDP to train RL policies using only a single demonstration per task generated from a model-based trajectory optimizer. Our approach incorporates an adaptive phase dynamics formulation to robustly track the demonstrations while accommodating dynamic uncertainties and external disturbances. We compare our method against prior motion imitation RL works and show that the learned policies achieve higher success rates across all considered tasks. These policies learn recovery maneuvers that are not present in the demonstration, such as re-grasping objects during execution or dealing with slippages. Finally, we successfully transfer the policies to a real robot, demonstrating the practical viability of our approach. For videos, please check: https://leggedrobotics.github.io/guided-rl-locoma/.

**Keywords:** Loco-Manipulation, Reinforcement Learning, Motion Imitation

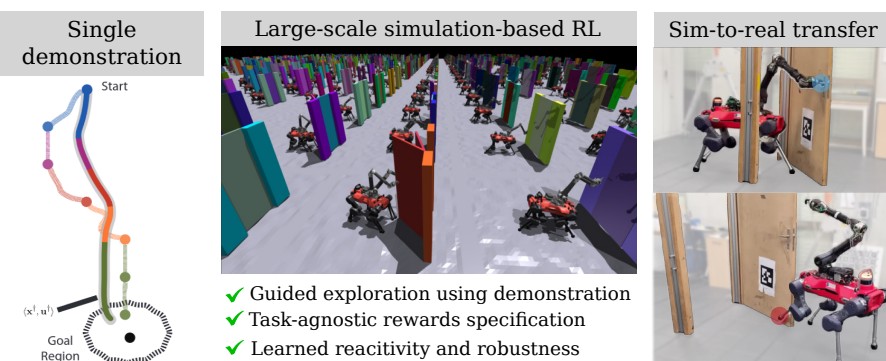

| Single demonstration | Large-scale simulation-based RL | Sim-to-real transfer |

✔ Guided exploration using demonstration
✔ Task-agnostic rewards specification
✔ Learned reactivity and robustness

Figure 1: A framework for learning loco-manipulation tasks, such as traversing a spring-loaded door and manipulating dishwashers. A single demonstration guides the RL training process to learn multi-contact behaviors (such as using the feet or the arm for interaction) without task-specific handcrafted rewards.

## 1 Introduction

Model-based optimal control techniques, such as trajectory optimization (TO) and model-predictive control (MPC), are valued for their inherent versatility. They can generate a range of easily interpretable behaviors, facilitating intuitive adjustments to the problem formulation [1, 2, 3]. However, these methods are sensitive to modeling mismatches and violation of assumptions. Conversely, reinforcement learning (RL) has demonstrated remarkable success in developing robust control policies for various contact-rich tasks, such as legged locomotion [4, 5], loco-manipulation [6, 7], and dexterous manipulation [8, 9]. Despite this, RL suffers from high sample inefficiency, the emergence of

---

\*These authors contributed equally. Their names are listed in alphabetical order.

8th Conference on Robot Learning (CoRL 2024), Munich, Germany.

unnatural behaviors, and the need for labor-intensive reward design. In this work, we aim to leverage the complementary strengths of TO and RL to mitigate their weaknesses.

We build upon the versatile TO-based framework from Sleiman et al. [10], which demonstrates the discovery of long-horizon multi-modal behaviors for poly-articulated systems. This framework can safely and reliably solve complex loco-manipulation tasks, accounting for the multiple interaction regions in the object (handle, surfaces) and the robot (individual limbs). These behaviors are challenging to achieve with pure RL, even with an extensive task-specific design process. However, the successful execution of TO-based behaviors depends heavily on reasonably accurate environment models and minimal external disturbances. This reliance is due to the underlying MPC-based controller's strict adherence to tracking open-loop references without the ability to replan or respond to significant deviations, such as slippages. This limitation of the MPC controller motivates our work: using RL to obtain policies for the robust execution of contact-rich interaction plans.

In this paper, we propose a Markov Decision Process (MDP) to efficiently train robust loco-manipulation policies, particularly for articulated object interaction, with task-agnostic rewards and hyperparameter tuning. Our approach uses dynamically feasible demonstrations generated from a TO-based framework [10] to guide the RL agent in learning complex behaviors. By maintaining consistent MDP parameters and utilizing only one demonstration per task, we present an approach to train control policies that track the generated demonstrations while handling modeling uncertainties, external disturbances, and unforeseen events such as handle slippage. We benchmark our approach against prior motion imitation works on four loco-manipulation tasks: door pushing and pulling and dishwasher opening and closing. Finally, we deploy the trained policies on a quadrupedal mobile manipulator, showcasing their robustness to unknown object models and reactivity to slippages.

## 2   Related Work

Imitation learning has been successfully applied to various manipulation tasks through its algorithmic variants such as behavioral cloning (BC) [11], demo-augmented policy gradient [12, 13] (which bootstraps the RL policy through BC), and more recently, generative models [14, 15]. However, these approaches often struggle with distributional shifts and require substantial training data, posing challenges for high-dimensional robotic systems.

An alternate class of methods relies on state-only references to guide an RL agent toward desired behaviors. This approach commonly involves conditioning a low-level policy on references obtained either offline from a motion library or online by a separate high-level module. For instance, Bergamin et al. [16] incorporates a motion matcher that selects and blends the best-fitting motion clip from an extensive MoCap database based on handcrafted features. The works from Kang et al. [17] and Jenelten et al. [18] update the policy with on-demand "optimal" references that are computed in an MPC-like fashion. While MPC can provide precise guidance, its high computational demands can significantly slow down the RL training process.

Another common approach is inverse RL, where a reward function is defined through the demonstrations. Adversarial motion priors (AMP) [19, 20, 21] formulate a *style* reward that is maximized when the learned policy yields state transitions similar to those in the dataset. Subsequent variants [22, 23, 24] propose a hierarchical architecture to address the mode-collapse issue in adversarial learning setups. However, these methods still require task-specific goal-directing rewards and a large amount of data to learn the style reward. In contrast to AMP-style approaches, *motion-imitation* methods guide RL policies to robustly track target trajectories directly [25, 26, 27, 28]. They use reward terms that encourage adherence to the demonstrations while allowing exploration around the reference motions.

Our work falls into the category of *motion-imitation* RL without any task-specific objectives. It closely relates to the work from Fuchioka et al. [26] and Bogdanovic et al. [27], as we aim to directly track and robustly stabilize offline trajectories generated using a fast trajectory optimizer. Differently from these works, we introduce domain-specific considerations relevant to multi-contact loco-manipulation tasks. Moreover, we propose a task-independent MDP formulation based on an

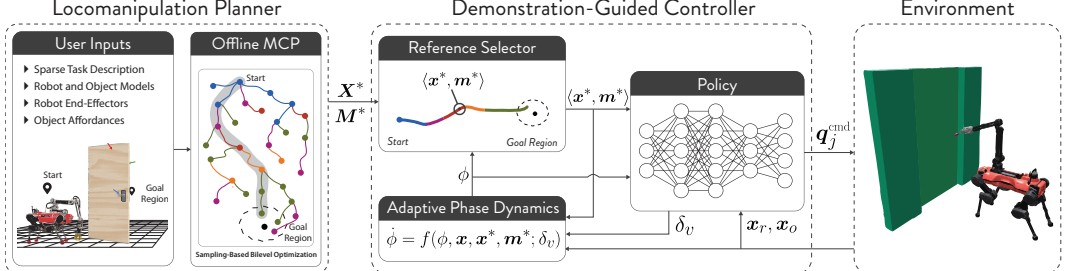

Figure 2: The **loco-manipulation planner** [10] generates references in the form of multi-modal plans consisting of continuous trajectories $\mathbf{X}^*$ and manipulation schedules $\mathbf{M}^*$. These are used by the **demonstration-guided controller** to select $\mathbf{x}^*$ and $\mathbf{m}^*$ adaptively based on the task phase $\phi$ and track them robustly. The controller receives full-state feedback and sends joint position commands to the robot.

adaptive phase dynamics model, which is crucial for successful task execution in the presence of large modeling uncertainties and external disturbances.

## 3 Approach

Our proposed approach consists of two steps, as illustrated in Fig. 2. First, we employ the planner from Sleiman et al. [10] to generate whole-body multi-contact behaviors for various loco-manipulation tasks. Subsequently, we train a neural network using RL to reliably track these behaviors while leveraging only one pre-computed trajectory per task as an "expert demonstration". We train this policy entirely in simulation with domain randomization to achieve a successful transfer to the real robot. At deployment, we assume access to the object's state. While this assumption limits the applicability in the wild, our primary focus is devising an MDP for learning robust behaviors across different loco-manipulation scenarios without requiring task-specific handcrafting.

### 3.1 Problem Formulation

We generate a single demonstration for each task using the multi-contact planner (MCP) from [10]. This planner takes the task specification and a set of user-defined object affordances (*e.g.* handle or surfaces) and the robot's end-effectors (*e.g.* the tool on the arm or the feet) for interaction as inputs. It then searches for possible robot-object interactions to provide a physically consistent demonstration based on a nominal robot and object model.

The demonstrations from the planner consist of the continuous robot and object state references $\mathbf{X}^* = \{\mathbf{x}_t^*\}_{t=1}^{T_{\text{task}}}$ and the manipulation schedule $\mathbf{M}^* = \{\mathbf{m}_t^*\}_{t=1}^{T_{\text{task}}}$ with $T_{\text{task}}$ being the demonstration's duration. In the case of articulated object interaction, the state reference $\mathbf{x}_t^*$ includes the robot's base pose $({}_I\mathbf{r}_{IB}^*, \mathbf{\Phi}_{IB}^*)$ in a fixed inertial frame $I$, the robot's joint positions $\mathbf{q}_j^*$, and the object's joint positions $\mathbf{q}_o^*$. The contact mode $\mathbf{m}_t^*$ specifies the interaction type for each end-effector (none, prehensile, or non-prehensile) and the object contacts involved at that timestep. Since the demonstrations for each task have varying lengths, we encode the notion of time into the task-phase $\phi \in [0, 1]$, where $\phi = 0$ and $\phi = 1$ imply the start and end of the demonstration, respectively [25].

The objective of the learning agent is to track these references while staying robust against variations in the kinematic and dynamic properties of the object, external disturbances, and unforeseen events such as slippages. In the naïvest way, one could increase the task phase linearly with time and track the corresponding references [25, 26]. However, as we show later in Sec. 5.1, this approach limits the success rate during object manipulation. For instance, if the door handle slips away and the door closes, strictly following the references will cause the robot to get stuck behind the door. Instead, we want to give the robot the time to recover and, thus, grow the task phase adaptively.

### 3.2 Adaptive Task Phase Dynamics

Linearly evolving the task phase with time (also referred to in this paper as the "nominal" formulation) can be written as: $\bar{\phi}_{t+1} = \bar{\phi}_t + \frac{1}{T_{\text{task}}} \cdot dt$, where $\bar{\phi}$ denotes the nominal phase and $dt$ is the environment time-step. As motivated earlier, we seek an adaptive mechanism that adjusts the phase $\phi$ depending on the current robot and object states. To this end, we propose the phase dy-

namics: $\phi_{t+1} = \phi_t + \frac{v_t^\phi}{T_{task}} \cdot dt$, where the task phase rate $v_t^\phi = \hat{v}_t^\phi + \sigma_1 \cdot \delta_{v,t}$ is determined by a state-dependent reference $\hat{v}^\phi := \hat{v}^\phi(\boldsymbol{x}, \boldsymbol{x}^*, \boldsymbol{m}^*)$ and a learnable residual term $\delta_v$ (scaled by $\sigma_1 > 0$). We opt for a reward-dependent $\hat{v}^\phi$ that reflects the task-level tracking accuracy:

$$\hat{v}_t^\phi(\boldsymbol{x}, \boldsymbol{x}^*, \boldsymbol{m}^*) = \prod_{i=1}^{L} \min\left(k \cdot \exp(-\lambda \cdot r_{i,t}(\boldsymbol{x}, \boldsymbol{x}^*, \boldsymbol{m}^*)), 1\right), \tag{1}$$

where $k > 1$ and $\lambda < 0$ are fixed constants, and $L$ is the number of task-level rewards $r_{i,t}$, with $r_{i,t} \leq 0$ at all time-steps. For articulated object interaction, $L = 3$ with terms corresponding to the tracking rewards for the robot's base pose and the object's joint positions, as specified in Sec. 3.3.

**Intuition behind $\hat{v}^\phi$:** The term $\hat{v}^\phi$ effectively pauses the phase evolution for large deviations from the current reference. As the tracking improves, it gradually approaches the nominal phase rate ($\to 1$). Since adhering strictly to the reference may not always be the best strategy to complete the task, we introduce a clipping in Eq. 1 to encode perfect tracking only within a certain margin defined by $k$ and $\lambda$. This operation is particularly useful when handling discrepancies between the nominal model used by the planner and the real model employed during training and deployment. The resulting task phase dynamics with $\hat{v}^\phi$ induces a curriculum-like effect during training. At the start of the training, when the tracking is poor, it helps the agent learn to recover to a single reference state. Eventually, as the agent improves, it learns to follow more of the demonstration.

**Intuition behind $\delta_v$:** In some instances, unforeseen slippage or large disturbances could render the object uncontrollable due to a complete loss of contact, resulting in significant deviations from the reference pose. In these situations, the term $\hat{v}^\phi \to 0$, and the robot cannot recover. To enable motion recovery in such scenarios, we introduce a residual phase $\delta_v$ that allows potentially speeding up, slowing down, and even decreasing the phase whenever necessary. This residual phase is outputted from the policy, allowing it to adapt to the task phase dynamics via learnable parameters.

### 3.3 Rewards

To design task-agnostic reward functions such that loco-manipulation behaviors in the generated demonstrations can be learned through the same MDP formulation, we split the rewards into three parts: reference-tracking rewards, task progress rewards, and penalties for smooth motions.

**Reference-tracking reward:** These are simply defined through the tracking errors between the current state $\boldsymbol{x}$ and the reference state $\boldsymbol{x}^*$ from the demonstration:

$$r^{track} = w_1 \cdot ||_I\boldsymbol{r}_{IB} - {}_I\boldsymbol{r}_{IB}^*||^2 + w_2 \cdot ||\boldsymbol{\Phi}_{IB} \boxminus \boldsymbol{\Phi}_{IB}^*||^2 + w_3 \cdot ||\boldsymbol{q}_j - \boldsymbol{q}_j^*||^2 \tag{2}$$
$$+ w_4 \cdot \mathbb{1}_{object}^* \cdot ||\boldsymbol{q}_o - \boldsymbol{q}_o^*||^2 + w_5 \cdot \mathbb{1}_{prehensile}^* \cdot ||_I\boldsymbol{r}_{IE} - {}_I\boldsymbol{r}_{IH}||^2,$$

where $w_i < 0$, $H$ is a non-fixed handle frame that is initialized at $I$ and moves with the object, and $\mathbb{1}_{...}^* := \mathbb{1}_{...}(\boldsymbol{m}^*)$ denotes indicator functions that depend on the reference manipulation mode $\boldsymbol{m}*$.

The indicator functions help avoid incurring irrelevant penalties that do not coincide with the behavior's mode schedule. The function $\mathbb{1}_{object}^*$ is true only when at least one end-effector is either already in contact or is establishing contact with the object. Essentially, it activates the object tracking error only when the object's state is controllable and essential for the task completion. Meanwhile, $\mathbb{1}_{prehensile}^*$ is true when a prehensile interaction is active. This choice helps improve the accuracy of prehensile contact and is particularly useful when we randomize the object models during training, as the location of the object frame $H$ varies.

**Task progress reward:** This reward term encourages the learning agent to progress in the task. It is defined as: $r^\phi = \kappa_1 \cdot \left[ \hat{v}^\phi \cdot \exp(-\kappa_2 \cdot ||\phi - \bar{\phi}||^2) \right]$, where $\kappa_1, \kappa_2 > 0$, and $\hat{v}^\phi$ and $\bar{\phi}$ are the reference task-phase rate and the nominal phase from Sec. 3.2. When the tracking is poor, $\hat{v}^\phi \to 0$ and the agent gets less reward for the task progress, i.e., $r^\phi \to 0$. Consequently, the agent gets encouraged to maximize the reference tracking reward and minimize the tracking deviation.

**Penalties:** These terms penalize the robot's joint accelerations and torques, base velocity in undesired directions, and abrupt action changes. Since these are standard penalties used in RL, we skip detailing them here for space reasons and refer the reader to Appendix C.2 for further details.

### 3.4 Observation and Action Spaces

**Observations:** The observations comprise the tracking errors in the robot and object states, the positions and velocities of all end-effector frames participating in prehensile interactions, the previous action, and the task-phase parameters. For a legged mobile manipulator with only one prehensile end-effector $E$ on the arm, the observations can be denoted as $\boldsymbol{o}_t = (\boldsymbol{o}_e \;\; \boldsymbol{o}_v \;\; \boldsymbol{o}_p \;\; \boldsymbol{a}_{t-1} \;\; \boldsymbol{o}_\phi)$, with:

$$\boldsymbol{o}_e = \begin{bmatrix} {}_I\boldsymbol{r}_{IB} - {}_I\boldsymbol{r}_{IB}^* \\ \boldsymbol{\Phi}_{IB} \boxminus \boldsymbol{\Phi}_{IB}^* \\ \boldsymbol{q}_j - \boldsymbol{q}_j^* \\ \boldsymbol{q}_o - \boldsymbol{q}_o^* \end{bmatrix}, \boldsymbol{o}_v = \begin{bmatrix} {}_B\boldsymbol{v}_{IB} \\ {}_B\boldsymbol{\omega}_{IB} \\ \dot{\boldsymbol{q}}_j \\ \dot{\boldsymbol{q}}_o \end{bmatrix}, \boldsymbol{o}_p = \begin{bmatrix} {}_I\boldsymbol{r}_{IE} \\ {}_E\boldsymbol{v}_{IE} \end{bmatrix}, \boldsymbol{o}_\phi = \begin{bmatrix} \phi \\ \hat{v}^\phi \end{bmatrix}. \tag{3}$$

The choice of the observation terms $\boldsymbol{o}_e$ and $\boldsymbol{o}_v$ resemble that of a classical PD control law, where $\boldsymbol{o}_e$ captures the position tracking error and $\boldsymbol{o}_v$ is the velocity tracking error with zero velocity targets. Effectively, the learned policy is a more complex tracking controller around the demonstrations. Additionally, while including $\boldsymbol{o}_p$ is not essential, we notice that having it improves the training as it eliminates the need to learn the mapping from the robot's state to operational space quantities.

**Actions:** The actions $\boldsymbol{a}_t = (\boldsymbol{a}_{q_j,t}, \delta_{v,t})$ are interpreted as the residuals over the robot's reference joint positions $\boldsymbol{q}_{j,t}^*$ and the reference phase rate $\hat{v}_t^\phi$ from Sec. 3.2. The robot's actions are sent to its actuators as joint position commands: $\boldsymbol{q}_{j,t}^{cmd} = \boldsymbol{q}_{j,t}^* + \sigma_2 \cdot \boldsymbol{a}_{q_j,t}$, with $\sigma_2 > 0$. While the planner [10] also provides reference joint velocities, we observed that including them in feed-forward control resulted in poorer performance. This finding aligns with that from Fuchioka et al. [26].

### 3.5 Training Setup

**Initial Distribution:** At environment resets, we apply the reference state initialization (RSI) strategy from imitation-based RL setups [25, 26]. In RSI, we randomly sample the initial phase $\phi_{init} \in [0, 1]$ and spawn the robot and object at their corresponding reference states. However, differently from RSI, we randomize the robot configuration uniformly around the reference state when $\phi_{init} = 0$. This alleviates the need to always start the robot at the initial reference state at deployment and lets the agent learn the necessary recovery actions to stay near the demonstration.

The learning task is then to complete the remainder of the task from these varying initial states. This approach exposes the policy to regions of the state space that are relevant to the reference behavior and are hard to reach independently. Additionally, it removes the need for the episode length to be at least as long as the demonstration $T_{task}$.

**Domain Randomization (DR):** Unlike [10], where the MPC-based tracking controller's success relies on a reasonable knowledge of the environment, we aim to make the learned policy robust to modeling uncertainties and disturbances through DR. This involves varying the properties of the robot (such as friction and base mass), the kinematic and dynamic models of the object (such as the door dimensions and spring loading parameters), and adding regular external disturbances to the system. For further details, please refer to Appendix C.4

**Curriculum:** We introduce a curriculum that incrementally increases the external disturbances, observation noise, reward penalties, and initialization offsets. The curriculum level $l_{rand} \in [0, 1]$ is updated according to the following rule:

$$l_{rand}^{k+1} = \begin{cases} l_{rand}^k + 0.25 & \text{if } p_\phi > 0.95 \\ l_{rand}^k - 0.25 & \text{if } p_\phi < 0.75 \end{cases}, \text{ with } p_\phi = \frac{\phi - \phi_{init}}{\bar{\phi} - \phi_{init}}. \tag{4}$$

The term $p_\phi$ represents the progress of the task phase relative to the nominal one. As the agent becomes proficient in tracking the demonstrations, it progresses to a higher difficulty level with larger DR and penalties. Conversely, if its performance is poor, it moves to a lower level.

## 4 Experimental Setup

We validate our framework on a legged mobile manipulator consisting of the quadruped ANYmal-D [29] with a 6-DoF robotic arm [30]. We consider four tasks: **Door Push/Pull**: traversing a spring-loaded push or pull door, and **Dishwasher Open/Close**: opening or closing a heavy dishwasher.

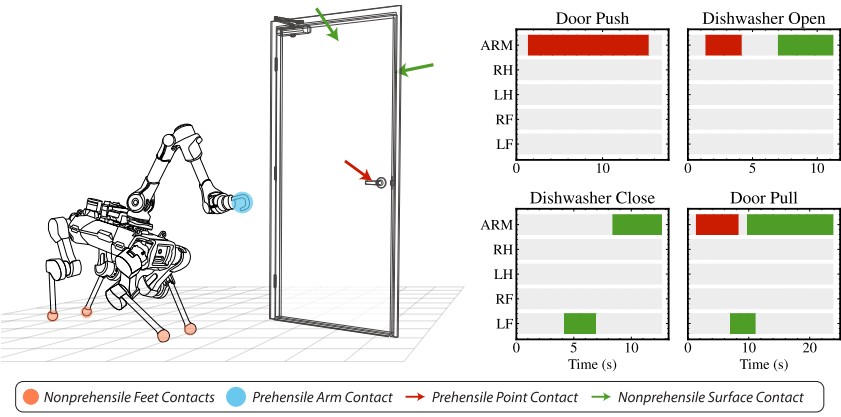

Figure 3: Manipulation schedules in the generated loco-manipulation demonstrations. For the quadrupedal mobile manipulator, ALMA, the end-effectors are: **ARM:** tool attached to a 6-DoF robotic arm, **LF:** left front foot, **RF:** right front foot, **LH:** left hind foot, and **RH:** right hind foot.

**Demonstration generation:** Fig. 3 illustrates the manipulation schedules $M^*$ of the generated demonstrations using [10]. Simpler tasks like door pushing are solvable with a single contact mode, while more complex tasks, such as door pulling, require sequencing multiple robot-object interaction modes. Navigating through a spring-loaded pull door stands out as the most complex task, both in terms of behavior discovery and execution time. For additional details, please check Appendix B.

**Large-scale RL training:** We simulate the training environment in NVIDIA Isaac Gym [31]. The simulator runs at $200\,\mathrm{Hz}$, and we decimate the policy to run at $50\,\mathrm{Hz}$. All policies are trained using the Proximal Policy Optimization (PPO) [32] algorithm. Training converges within 6 hours per task on a single NVIDIA RTX 4090Ti. Notably, we apply the same MDP formulation across all tasks with identical environment and agent hyperparameters. These are specified in Appendix C.

**Baselines:** Behavior cloning (BC) requires labeled actions and a large training dataset. Given that we operate with a few demonstrations that also lack action labels, we do not compare our method against BC. Instead, we evaluate our method against other motion-imitation RL formulations:

- **Nominal phase:** This formulation is based on existing works in motion-imitation RL [25, 26]. It uses the task phase dynamics with $v^\phi = 1$ and no learned residual phase rate.

- **Nearest Neighbor Search (NNS):** An alternative phase update mechanism inspired by the time-independent training stage of [27]. It aims to minimize the distance between the current system state and the closest reference in the demonstration dataset.

## 5 Results and Analysis

### 5.1 Simulation Experiments

**Comparisons:** We evaluate each learned policy in 4096 environments with randomized robot and object properties and random pushes. We report the success rates across the four tasks in Table 1. We observe that our proposed method performs significantly better than other motion-imitation RL setups, even without including the residual phase action $\delta_v$. This result highlights the significance of integrating the reward-driven adaptive phase dynamics. The nominal phase update can reliably accomplish the relatively straightforward task of navigating through push doors. However, it struggles with more complex tasks, achieving notably lower success rates for door pulling. Meanwhile, the NNS strategy fails across all the tasks. While experimenting with various training setups to incorporate NNS, we consistently found the phase stuck at certain points along the reference path. We hypothesize this occurs because the policy exploits the NNS update mechanism to achieve a state where the phase remains unchanged, thereby maximizing the reference tracking reward.

**Ablations:** To evaluate the need for demonstrations as whole-body trajectories, we omit the robot's joint-level references, relying on partial task-level references that pertain solely to the floating base

Table 1: Comparison and ablation study of different MDP formulations to train RL policies for the four loco-manipulation tasks. The average success rate is reported over 4096 episodes, with success defined as the successful execution of more than 95% of the demonstration.

| Category | Door Push | Dishwasher Open | Dishwasher Close | Door Pull |
|---|---|---|---|---|
| NNS-based $\phi$ | 00.00% | 00.00% | 00.00% | 00.00% |
| Nominal $\phi$ | 76.83% | 53.44% | 70.68% | 15.38% |
| Adaptive $\phi$ w/ Residual $\delta_v$ (**Ours**) | **98.36**% | **98.60**% | 96.46% | **96.33**% |
| Ours w/o Joint References | 97.17% | 00.00% | 00.00% | 00.00% |
| Ours w/o Curriculum | 97.69% | 95.25% | 97.38% | 95.46% |
| Ours w/o Residual $\delta_v$ | 96.80% | 97.22% | **97.68**% | 95.21% |

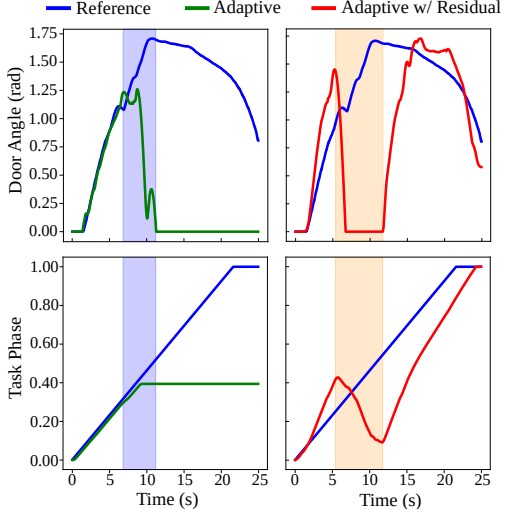

Figure 4: Trajectories for the door-pulling task involving a loss of contact (indicated by shaded regions). The top and bottom rows show the door and task phase trajectories.

Figure 5: Effect of randomization for the door pulling task. **Grey:** The DR region used during training. **Red:** The nominal parameter.

and the object trajectories. As shown in Table 1, only the door pushing remains solvable under this scheme. The reliance on the robot's foot contacts for manipulation necessitates whole-body guidance to solve the other tasks. Moreover, we see that the curriculum from Sec. 3.5 helps improve the agent's performance across most tasks. During training, we observe that the curriculum also enables faster convergence and smoother policies. Adding the residual phase rate only improves the performance slightly in these comparisons. However, as shown in Fig. 4 for the door-pulling task, relying solely on the reference $\hat{v}^\phi$ for the phase dynamics brings the task progression to a halt during an inadvertent loss of contact. In contrast, the learned residual action decreases the phase, allowing the robot to re-establish contact with the handle and successfully complete the task.

**Robustness Evaluation:** While the generated loco-manipulation plan is done for nominal parameters of the object, we train the RL policies with various randomizations, as mentioned in Sec. 3.5. To assess the robustness of the learned policies, we investigate their performance on different door parameters for the door-pulling task. We remove all DRs during these evaluations to ensure a controlled setting. For comparison, we train an RL policy with no DR as an alternative to the MPC-based controller in [10]. Based on Fig. 5, the policy trained only with nominal door properties fails quickly when the variations become too large. In contrast, the policy trained with random doors supports a wider range of door parameters. When the handle offset is significant, the policy trained with random doors displays a "search" behavior to find and grasp the handle. Meanwhile, the policy with a nominal door only works reliably well around the handle length. This analysis demonstrates that although the generated demonstrations are not for the simulated door models, the RL policy discovers behaviors around them that help it generalize to a large set of doors.

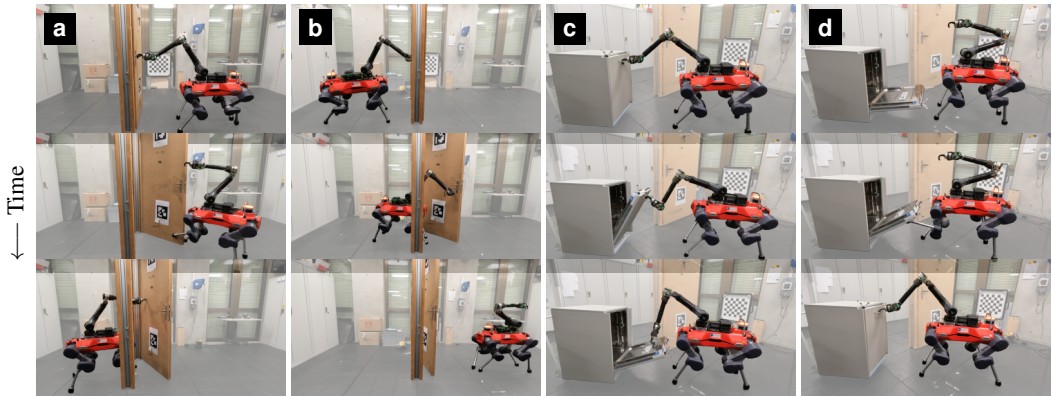

Figure 6: Hardware deployment for traversing spring-loaded (a) pull and (b) push doors and manipulating a heavy dishwasher to (c) open and (d) close it. For videos, please check the website.

**Multi-task Learning from Demonstrations**    We experiment with a multi-task training setup where a single policy jointly learns all four tasks. It takes a one-hot vector to indicate the desired demonstration to execute. We train the policy with the same hyperparameters and number of environments from before and obtain a reduced average success rate ($84\%$ instead of $97\%$). By simply doubling the number of environments during training (*i.e.*, collecting twice the number of samples), we observe that it achieves the performance of individual task-specific policies.

## 5.2    Real-World Deployment

We demonstrate the proposed approach on all four tasks on hardware. A motion capture system provides the handle location at the start and the joint angle for the door and the dishwasher. For the robot, we rely on its onboard state estimation. For each task, we perform six continuous runs, where we manually reset the object, place the robot randomly in front of the object, and execute the policy. As shown in Fig. 6, the deployed policies exhibit smooth motions and preserve the multi-contact behavior from the demonstration. They complete all the tasks six times in a row. Additionally, the policies yield different motions every time, which is particularly noticeable during traversing a pull door. They are able to handle random base placement, recover from slippages, and re-grasp the handle when it misses it. Please refer to the supplementary video to observe these behaviors.

## 5.3    Limitations

On hardware deployment, we notice that if the robot starts too far away or too close to the object, it aggressively tries to adjust its configuration. This, at times, led to the robot falling down. However, in practice, a navigation planner can appropriately place the robot in front of the object, so only local adjustments are needed, which the learned policies can handle. Additionally, while the policies can handle large intra-object category variations, they may still fail on certain object instances. For instance, if the door is too small, the policy fails to traverse through it as it collides with the door. In such cases, we need to use more demonstrations for the training. Lastly, a natural extension of this work is to devise perceptive loco-manipulation policies that rely on onboard sensing [5, 7].

## 6    Conclusion

Our work integrates model-based TO and model-free RL to develop robust tracking policies for multi-contact loco-manipulation tasks. Central to our approach is a task-agnostic MDP formulation that utilizes loco-manipulation demonstrations generated by a precise trajectory planner. Unlike previous imitation-based RL methods that depend on time-based nominal phase updates, we showed that our state-dependent adaptive phase dynamics facilitate successful task execution despite modeling inaccuracies and significant external disturbances. We validated our framework on a quadrupedal mobile manipulator performing four complex long-horizon tasks, such as navigating spring-loaded doors and manipulating heavy dishwashers. Additionally, we demonstrated that the learned policies transfer to hardware successfully and can effectively recover from slippages and missed handles, overcoming the limitations of the MPC-based tracking controller from [10].

**Acknowledgments**

This research was supported by the European Union's Horizon Europe Framework Programme under grant agreements No 101070596, 101121321, and 852044.

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

# A    Notations / Symbols

Table 2: Definition of Symbols

| Symbol | Description |
|---|---|
| $g^*$ | Value of the quantity $g$ from the demonstration |
| $\bar{g}$ | Value of the quantity $g$ from the nominal phase formulation |
| $_A \boldsymbol{r}_{AB} \in \mathbb{R}^3$ | Position of frame B from frame A in frame A |
| $\boldsymbol{\Phi}_{AB} \in SO(3)$ | Orientation of frame B in frame A |
| $_A \boldsymbol{v}_{AB} \in \mathbb{R}^3$ | Linear velocity of frame B from frame A in frame A |
| $_A \boldsymbol{\omega}_{AB} \in \mathbb{R}^3$ | Angular velocity of frame B from frame A in frame A |
| $\boxminus : SO(3) \times SO(3) \to \mathbb{R}^3$ | Difference between two orientations as a rotation error vector [33] |
| $B$ | Frame attached to the base of the robot |
| $E$ | Frame attached to the arm's end-effector of the robot |
| $H$ | Frame attached to the handle's center on the object |
| $I$ | Inertial frame (fixed to the initial frame $H$ on execution) |
| $\boldsymbol{h}_{\text{com}}$ | Centroidal momentum of the robot |
| $\boldsymbol{q}_b = (_I \boldsymbol{r}_{IB}, \boldsymbol{\Phi}_{IB})$ | Base pose of the robot |
| $\boldsymbol{q}_j$ | Joint positions of the robot |
| $\boldsymbol{q}_o$ | Joint positions of the object |
| $\dot{\boldsymbol{q}}_j$ | Joint velocities of the robot |
| $\dot{\boldsymbol{q}}_o$ | Joint velocities of the object |
| $\ddot{\boldsymbol{q}}_j$ | Joint acceleration of the robot |
| $\boldsymbol{\tau}_j$ | Joint torques applied to the robot |
| $\boldsymbol{m}$ | Manipulation contact mode between robot and the object |
| $\boldsymbol{x}_r = (\boldsymbol{h}_{\text{com}, I} \boldsymbol{r}_{IB}, \boldsymbol{\Phi}_{IB}, \boldsymbol{q}_j)$ | Kinematic state of the robot |
| $\boldsymbol{x}_o = (\boldsymbol{q}_o, \dot{\boldsymbol{q}}_o)$ | Kinematic state of the object |
| $\boldsymbol{x} = (\boldsymbol{x}_r, \boldsymbol{x}_o)$ | Kinematic state of the robot and the object |
| $\boldsymbol{M} = \{\boldsymbol{m}_t\}_{t=1}^{T_{\text{task}}}$ | Manipulation schedule (Sequence of manipulation contact modes) |
| $\boldsymbol{X} = \{\boldsymbol{x}_t\}_{t=1}^{T_{\text{task}}}$ | Kinematic state trajectory (Sequence of kinematic states) |
| $T_{\text{task}}$ | Duration (in s) of the trajectory for the task |
| $\mathbb{1}_{object} := \mathbb{1}_{object}(\boldsymbol{m})$ | True **iff** at least one end-effector is either already in contact with or is establishing contact with the object |
| $\mathbb{1}_{prehensile} := \mathbb{1}_{prehensile}(\boldsymbol{m})$ | True **iff** a prehensile interaction is active |
| $\phi \in [0, 1]$ | Phase signal that helps index the reference trajectory $(\boldsymbol{X}^*, \boldsymbol{M}^*)$ |
| $v_t^\phi$ | Task phase rate (first order-dynamics model for $\phi$) |
| $\hat{v}_t^\phi$ | Task phase rate based on reward-dependent functions |
| $\bar{\phi}$ | Nominal phase signal computed using $v^\phi = \frac{1}{T_{\text{task}}}$ |
| $\delta_v$ | Residual phase rate (output from the policy) |
| $dt$ | Step-size (in s) of the environment |
| $\boldsymbol{o}_t$ | Observation from the environment at time-step $t$ |
| $\boldsymbol{a}_t$ | Action applied to the environment at time-step $t$ |
| $r_t$ | Reward from the environment at time-step $t$ |

# B    Demonstrations from the Loco-Manipulation Planner

The loco-manipulation planner from Sleiman et al. [10] efficiently generates physically consistent demonstrations for our proposed framework. The planner relies on a high-fidelity model that integrates the robot's full centroidal dynamics and first-order kinematics with the object's full dynamics [34]. This helps ensure that the discovered behaviors are dynamically feasible. The planner outputs the following the continuous states and system inputs: $\boldsymbol{x} := (\boldsymbol{x}_r \ \boldsymbol{x}_o) = (\boldsymbol{h}_{com} \ \boldsymbol{q}_b \ \boldsymbol{q}_j \ \boldsymbol{q}_o \ \dot{\boldsymbol{q}}_o)$ and $\boldsymbol{u} = (\mathbf{w}_e \ \dot{\boldsymbol{q}}_j)$, where the robot state $\boldsymbol{x}_r$ includes the centroidal momentum $\boldsymbol{h}_{com}$, base pose $\boldsymbol{q}_b$, and joint positions $\boldsymbol{q}_j$, whereas the object state $\boldsymbol{x}_o$ consists of its generalized coordinates $\boldsymbol{q}_o$ and velocities $\dot{\boldsymbol{q}}_o$. The control input $\boldsymbol{u}$ is composed of the robot's joint velocities $\dot{\boldsymbol{q}}_j$ and the contact wrenches $\boldsymbol{w}_e$ acting at the robot's end-effectors.

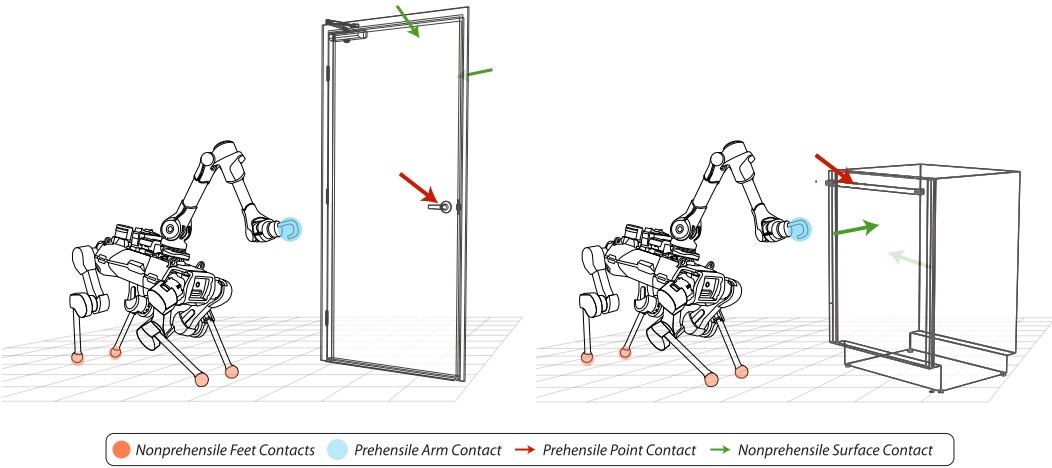

Figure 7: Illustration of the loco-manipulation tasks: i) traversal of a large articulated object, and ii) dishwasher manipulation. The user-defined robot end-effector contacts and object affordances are highlighted.

Moreover, from a set of user-defined object affordances and the robot's end-effectors for interaction, a discrete variable $m_k$ represents the manipulation contact mode. A contact mode is a state-action pair, where the contact state encodes possible robot-object interaction combinations, and a contact-switching action indicates whether a contact is established, broken, or maintained. By exploiting loco-manipulation-specific pruning rules, the planning algorithm in [10] efficiently solves for a multi-modal sequence via a sampling-based bi-level search over manipulation modes $m_k$ and continuous state-input trajectories $\langle x_k(t), u_k(t) \rangle$, aiming to connect the start and goal states. It then refines the resulting plan through a single long-horizon TO while fixing the discovered contact sequence. We refer the reader to [10] for further details on the multi-contact planner.

While the references generated from the planner contain the control inputs $u_k(t)$, these signals are usually tracked on hardware through a whole-body quadratic programming (QP) controller. This controller computes the necessary joint torques to achieve the desired motions. However, the QP controller's robustness is limited because of its several assumptions, such as no slippages and precise command tracking. We use only the reference states $X^*$ and contact modes $M^*$ to address these limitations and guide the RL training process. This approach allows the NN policy to learn the underlying actuator dynamics during training and adapt better to the inherent uncertainties and variations encountered during real-world operations.

Table 3 summarizes the single demonstrations generated for each task. In less than a minute, all demonstrations are discovered on an Intel Core i7-10750H CPU@2.6GHz hexacore processor. Navigating through a spring-loaded pull door stands out as the most complex task. This can be attributed to the long time horizon, the requirement for a stable prehensile interaction, and the multiple contact transitions involved. Discovering these modes using standard RL would necessitate carefully designing handcrafted rewards, which we want to alleviate through our formulation.

Table 3: Computation time and trajectory duration for demonstrations generated using the planner [10].

| Task | Computation Time (s) | Trajectory Duration (s) | Trajectory Length |
|---|---|---|---|
| Door Push | 6.8 | 16.8 | 1195 |
| Dishwasher Open | 25.0 | 11.2 | 814 |
| Dishwasher Close | 23.1 | 12.6 | 902 |
| Door Pull | 44.2 | 23.8 | 1725 |

## C   MDP Formulation and PPO Training

This section summarizes the terms that formulate the proposed MDP and the learning algorithm.

## C.1 Observation Terms

Table 4 specifies the observation terms and the noise added to them during training. The critic obtains the same observations as the actor but without any noise applied. Importantly, for the adaptive-phase dynamics formulation, the previous action $\boldsymbol{a}_{t-1}$ comprises both the robot commands $\bar{\boldsymbol{a}}_{t-1}$ and the residual phase $\delta_v$.

Table 4: Observation Terms Summary. We do not perform any scaling or clipping on individual observation terms. All noise models are additive in nature.

| Term Name | Definition | Noise |
|---|---|---|
| Robot Base Position Difference | $_I\boldsymbol{r}_{IB} - _I\boldsymbol{r}_{IB}^*$ | $\mathcal{U}(-0.05, 0.05)$ |
| Robot Base Orientation Difference | $\boldsymbol{\Phi}_{IB} \boxminus \boldsymbol{\Phi}_{IB}^*$ | $\mathcal{U}(-0.1, 0.1)$ |
| Robot Base Linear Velocity | $_B\boldsymbol{v}_{IB}$ | $\mathcal{U}(-0.1, 0.1)$ |
| Robot Base Angular Velocity | $_B\boldsymbol{\omega}_{IB}$ | $\mathcal{U}(-0.2, 0.2)$ |
| Robot Joint Position Difference | $\boldsymbol{q}_j - \boldsymbol{q}_j^*$ | $\mathcal{U}(-0.01, 0.01)$ |
| Robot Joint Velocity | $\dot{\boldsymbol{q}}_j$ | $\mathcal{U}(-1.5, 1.5)$ |
| Robot Arm End-effector Position | $_I\boldsymbol{r}_{IE}$ | $\mathcal{U}(-0.05, 0.05)$ |
| Object Joint Position Difference | $\boldsymbol{q}_o - \boldsymbol{q}_o^*$ | $\mathcal{U}(-0.05, 0.05)$ |
| Nominal Task Phase | $\bar{\phi}$ | - |
| Adaptive Task Phase | $\phi$ | $\mathcal{U}(-0.005, 0.005)$ |
| Adaptive Task Phase Speed | $\hat{v}^\phi$ | $\mathcal{U}(-0.005, 0.005)$ |
| Previous Action | $\boldsymbol{a}_{t-1}$ | - |

## C.2 Reward Terms

To design a task-agnostic reward function, we split the reward function into generic reference-tracking terms that stabilize the open-loop trajectories and standard penalty terms that ensure smooth motions: $\bar{r}^{total} = r^{track} + l_{rand} \cdot r^{regularize}$. For the adaptive phase formulation, the reward also includes the task progress: $r^{total} = \bar{r}^{total} + r^\phi$. The individual terms are:

$$
\begin{aligned}
r^{track} &= w_1 \cdot ||_I\boldsymbol{r}_{IB} - _I\boldsymbol{r}_{IB}^*||^2 + w_2 \cdot ||\boldsymbol{\Phi}_{IB} \boxminus \boldsymbol{\Phi}_{IB}^*||^2 + w_3 \cdot ||\boldsymbol{q}_j - \boldsymbol{q}_j^*||^2 \\
&\quad + w_4 \cdot \mathbb{1}_{object}^* \cdot ||\boldsymbol{q}_o - \boldsymbol{q}_o^*||^2 + w_5 \cdot \mathbb{1}_{prehensile}^* \cdot ||_I\boldsymbol{r}_{IE} - _I\boldsymbol{r}_{IH}||^2, \\
r^\phi &= \kappa_1 \cdot \left[ \hat{v}^\phi \cdot \exp(-\kappa_2 \cdot ||\phi - \bar{\phi}||^2) \right], \\
r^{regularize} &= \beta_1 \cdot ||\boldsymbol{\tau}||^2 + \beta_2 \cdot ||\dot{\boldsymbol{v}}_j||^2 + \beta_3 \cdot ||\boldsymbol{v}_j||^2 + \beta_4 \cdot ||_B\boldsymbol{v}_{IB}^z||^2 + \beta_5 \cdot ||_B\boldsymbol{\omega}_{IB}^{xy}||^2 \\
&\quad + \beta_6 \cdot ||\boldsymbol{a}_t - \boldsymbol{a}_{t-1}||^2,
\end{aligned}
$$

where symbols have their meanings from Table 2. The corresponding weights are in Table 5.

## C.3 Initial-State Distribution

We apply additive offsets to the robot's reference configuration at $\phi_{init} = 0$ so that the policy is robust to varying initial locations of the robot in front of the door. Ideally, we would like to apply this at any randomly sampled $\phi$; doing so is non-trivial due to the difficulty in filtering invalid collision configurations.

## C.4 Domain Randomization

Domain randomization helps mitigate overfitting to specific models and addresses inherent unmodelled effects by introducing variability during training. In our setup, it takes the following form:

- **Object's kinematics**: These include object dimensions (*e.g.* door width and height), positioning of object affordances (*e.g.* handle location on the panel), and handle types (cylinder or box). For every object category, we load 128 different kinematic variations.

- **Object's dynamics**: These include friction and restitution, spring-damping coefficients for the hinge joint, and constant force/torque offset on the hinge joint.

Table 5: Reward Terms Summary. The environment scales the reward weights with the time-step $dt$ [35]. For brevity, we drop the time-step $t$ from individual quantities unless necessary. We use the same reward weights for all the loco-manipulation tasks.

| Term Name | Definition | Weight |
|---|---|---|
| Robot Base Position Tracking | $\|_I \boldsymbol{r}_{IB} - _I \boldsymbol{r}_{IB}^*\|^2$ | $-0.2$ |
| Robot Base Orientation Tracking | $\|\boldsymbol{\Phi}_{IB} \boxminus \boldsymbol{\Phi}_{IB}^*\|^2$ | $-0.2$ |
| Robot Joint Position Tracking | $\|\boldsymbol{q}_j - \boldsymbol{q}_j^*\|^2$ | $-0.2$ |
| Object Joint Position Tracking | $\mathbb{1}_{object}^* \cdot \|\boldsymbol{q}_o - \boldsymbol{q}_o^*\|^2$ | $-0.2$ |
| Robot Arm End-effector Position Tracking | $\mathbb{1}_{prehensile}^* \cdot \|_I \boldsymbol{r}_{IE} - _I \boldsymbol{r}_{IH}\|^2$ | $-10.0$ |
| Action Rate | $\|\boldsymbol{a}_t - \boldsymbol{a}_{t-1}\|^2$ | $-0.05$ |
| Robot Base Linear Velocity (along $z$) | $\|_B \boldsymbol{v}_{IB}^z\|^2$ | $-0.5$ |
| Robot Base Angular Velocity (along $xy$) | $\|_B \boldsymbol{\omega}_{IB}^{xy}\|^2$ | $-0.05$ |
| Robot Joint Velocity | $\|\dot{\boldsymbol{q}}_j\|^2$ | $-1.0 \times 10^{-5}$ |
| Robot Joint Acceleration | $\|\ddot{\boldsymbol{q}}_j\|^2$ | $-1.0 \times 10^{-5}$ |
| Robot Applied Joint Torque | $\|\boldsymbol{\tau}_j\|^2$ | $-2.5 \times 10^{-5}$ |
| Task Progress (only with adaptive phase) | $\hat{v}^\phi \cdot \exp(-10.0 \cdot \|\phi - \bar{\phi}\|^2)$ | $25.0$ |

- **Robot's dynamics**: Similar to object dynamics, we vary the friction and restitution within $[0.4, 1.0]$. Additionally, the mass of the robot's base is randomized within $\pm 10\%$ of its nominal values.

- **External disturbances**: At randomly sampled episode intervals, external pushes are applied to both the robot and the object. For the robot, this implies adding random velocity (pushes) to the robot's base. For the door, this is done by applying a randomly sampled external force on the door panel.

## C.5 Termination Term

We trigger episode termination when the robotic system loses balance or episode length times out. Typically, we infer a fall from a significant force acting on the robot's base, indicating ground contact. However, distinguishing the source of this force becomes challenging during loco-manipulation tasks, as contact between the robot's base and an object is both expected and sometimes permissible. Thus, we rely on the base to not drop below $0.3\,\mathrm{m}$ to correctly detect falls.

## C.6 Adaptive-Phase Hyperparameters

This section lists the hyperparameters for the remainder of the MDP formulation. These include parameters for scaling the input actions and those for the adaptive-phase formulation in Eq. 1.

Table 6: MDP Hyperparameters.

| Hyperparameter | Value |
|---|---|
| Episode length | $15\,\mathrm{s}$ |
| Simulation time-step | $0.005\,\mathrm{s}$ |
| Control decimation | 4 |
| Residual phase action scale: $\sigma_1$ | 0.01 |
| Robot action scale: $\sigma_2$ | 0.5 |
| Adaptive phase (1): $w_1$ | -0.2 |
| Adaptive phase (1): $w_2$ | -0.2 |
| Adaptive phase (1): $w_4$ | -0.2 |
| Adaptive phase (1): $\lambda$ | -50.0 |
| Adaptive phase (1): $k$ | 10.0 |

## C.7 Learning Algorithm

For each task, we train the policy using the on-policy RL algorithm, Proximal Policy Optimization (PPO) [32]. The actor and critic networks are designed as a Multi-Layer Perceptron (MLP) with

a $[256 \times 128 \times 64]$ hidden-layer structure and an *ELU* activation function. A complete list of hyperparameters and their values is specified in Table 7.

Table 7: PPO Hyperparameters

| Hyperparameter | Value |
|---|---|
| Empirical Normalization | True |
| Learning Rate (start of training) | 1e-3 |
| Learning Rate Schedule | "adaptive" (based on KL-divergence [35]) |
| Discount Factor | 0.99 |
| GAE Discount Factor | 0.95 |
| Desired KL-divergence | 0.01 |
| Clip Range | 0.2 |
| Entropy Coefficient | 0.0 |
| Value Function Loss Coefficient | 1.0 |
| Batch Size | $245{,}760 \ (4096 \times 60)$ |
| Mini-Batch Size | $61{,}440 \ (4096 \times 15)$ |
| Number of Epochs | 5 |
| Number of iterations | 10,000 |

## D  Supplementary Discussions

### D.1  Object Locking Mechanism

Our policy is trained on doors where handles are treated as fixed object-attached links. However, a typical door can only be opened after being unlocked using its handle. By adapting our environment to incorporate a door-locking mechanism, our current setup results in behaviors involving the robot pushing the handle downwards, but with low success rates. A possible way to resolve this issue is using an asymmetric actor-critic structure, where the critic obtains the handle angle.

### D.2  Unknown Object Type

In real-world scenarios, we expect the robot to autonomously traverse diverse doors without specifying the type of door (*i.e.*, push or pull door). One way to achieve this objective would be to first train a multi-task policy that can execute both door-traversal behaviors and then separately train a door-type estimator that outputs the appropriate task command to the policy.

### D.3  Unknown Object State

In the current hardware experiments, we rely on explicit sensors such as door encoders to obtain the joint position of the panel. However, in more realistic scenarios, this information needs to be extracted from the robot's onboard sensors. One mechanism to achieve this goal is by leveraging teacher-student training and treating the current policy as the teacher policy [5].

### D.4  Applicability to Other Scenarios

In this work, we demonstrated the approach to multi-contact tasks that primarily involved articulated object manipulation. However, there is an open question on the generalization of the approach to other tasks and robots. For tasks where a single demonstration is sufficient to solve the task, we believe our method should work in these scenarios. However, for tasks where a complete re-planning is necessary (for instance, the rearrangement of objects), training an online replanner becomes necessary to provide new references for the policy to track. We leave this as part of our future work.

