# OpenReview forum: "Guided Reinforcement Learning for Robust Multi-Contact Loco-Manipulation"
_robot-learning.org/CoRL/2024/Conference — CoRL 2024_

### Official Review · Reviewer_1Sf9 · 2024-07-18
**Guided Reinforcement Learning with adaptive phase dynamics offer a promising approach to handling contact-rich loco-manipulation tasks**

**Originality:** 3
**Technical Quality:** 3
**Clarity Of Presentation:** 3
**Potential Impact:** 3
**Recommendation:** 3
**Confidence:** 4

**Review:**

Strengths:
1. The paper is well motivated and easy to follow.
2. The ablations are carefully designed to justify each design component, including the adaptive task phase variable, joint references and curriculum learning.
3. Both simulation and hardware experiments show promising performance against disturbances on multi-contact loco-manipulation tasks., using only a single demonstration per task.

Weaknesses:
1. Lack of Comparison with Baselines: the paper does not compared with other baselines such as vanilla IL or RL from scratch to validate the effectiveness of imitation-based RL.
2. Uncommon MDP Formulation Designs: some of the formulation designs are not common in MDP (e.g., the task phase variable, residual action). The paper would benefit from a high-level intuition and clear explanations of these designs to highlight how they differ from existing formulations and why they are necessary.
3. Limited Real-World Experiments: while the simulation includes the dishwasher task, the hardware experiments are only limited to pushing or pull the door (without actually traversing through the door).

**Quality Of The Limitations Section:**

3

**Questions For Rebuttal:**

1. Could you explain the physical meaning and effect of the residual action $\delta_v$ more clearly?
2. Could you give an overview of the proposed formulations and what are their advantages over existing ones?
3. Although the NNS results in 0% success rate, it could be informative to visualize their trajectories as well. Are they stuck at the onset of unforeseen slippage and large disturbances?

**Robotics Focus:**

4

**Summary Of Paper:**

The paper proposes a generic MDP framework that includes a task phase variable to train loco-manipulation RL policies, guided by trajectory optimization.

**Summary Of Recommendation:**

I recommend the acceptance of this paper. The proposed MDP framework for guided RL and adaptive phase dynamics offer a promising approach to handling complex, contact-rich tasks. However, including comparisons with baseline methods and providing clearer intuitions for the unique design components would strengthen the paper further.

---

### Official Review · Reviewer_kZSQ · 2024-07-22
**Promising work but needs further improvement**

**Originality:** 2
**Technical Quality:** 2
**Clarity Of Presentation:** 3
**Potential Impact:** 2
**Recommendation:** 2
**Confidence:** 3

**Review:**

This paper proposes a method to address four challenging tasks and shows nice results in simulation. The approach is also partially demonstrated on a real robot system. Ablations validate the efficacy of various components, and domain randomization during training improves the robustness of the behavior.

However, I have concerns about the generality of the approach and the real robot implementation. The authors describe a goal of devising a generic formulation that works across scenarios without task-specific handcrafting. However, the lack of breadth in evaluation tasks and the complexity of the approach do not support the claim of a method that is general. This paper reads to me as a specific instantiation of a hierarchical-RL-based approach for these specific tasks.

For the real robot evaluation, the video is rather confusing. The robot does not complete the task, it just opens the door and then video stops there, the robot does not walk through the door. Also, the real robot is completely stationary during most of the video. What is happening during those long pauses? Did the code crash?

Another weakness of the real world implementation is the assumption of access to ground truth states. The authors implement this by installing an encoder on the door hinge and putting fidicial markers on the door surface. This limits the practicality of the approach if the ultimate goal is deployment in the real world.

**Quality Of The Limitations Section:**

3

**Questions For Rebuttal:**

Presentation
- The message of the paper can be refined, perhaps not claiming a general method if it cannot be backed up
- Some of the terminology in the paper is not widely used in RL and yet are not introduced or defined anywhere. Here are a few examples: references, schedule, phase.

Evaluation:
- There are ablations performed on the method itself but there does not seem to be a comparison to a widely-used existing approach as a baseline

Real robot:
- The robot should complete the task or at least the authors should explain why it is only able perform half of the task
- If the long pauses are due to code implementation, that should be addressed

Minor issues:
- Line 20: "has shown demonstrated" -> "has demonstrated"
- Line 53: "behaviors.This" -> "behaviors. This"

**Robotics Focus:**

4

**Summary Of Paper:**

This paper proposes a method to enable a quadruped loco-manipulator to navigate doors and open/close dishwashers. The approach first uses a planner to generate a trajectory of high-level waypoints to accomplish the task. These waypoints are sent to a controller, which keeps track of progress towards the goal along the trajectory, and sends intermediate subgoals to the low-level policy. The policy takes in a target state and manipulation mode as input, and is trained using PPO. The approach is demonstrated in both simulation and on a real robot.

**Summary Of Recommendation:**

This work is promising but needs improvements in presentation, comparison to another method, and real robot implementation

---

### Official Review · Reviewer_9jLh · 2024-07-22
**New formulation for updating phase enables more robust loco-manipulation for door opening**

**Originality:** 3
**Technical Quality:** 3
**Clarity Of Presentation:** 3
**Potential Impact:** 2
**Recommendation:** 3
**Confidence:** 3

**Review:**

This work claims to introduce a task-independent MDP formulation, based on an adaptive phase dynamics model, that allows the RL agent to learn robust policies guided by a single demonstration.

It has several strengths:
1. The adaptive phase dynamics formulation is an interesting idea with demonstrated effectiveness in robustly solving door opening tasks.
2. This contribution reduces the burden of manual tuning in sim2real RL, in that it reuses the same MDP formulation and algorithm
hyperparameters for solving a suite of tasks, which is an important goal in machine learning research for robotics.
3. The task of opening a spring-loaded door is sufficiently hard, and the real-world behavior demonstrated in the supplemental video looks convincingly robust, to motivate the author's proposed approach.

It also has several weaknesses:
1. The organization of the methods section is confusing, in that the equations that describe the formulation the authors actually use are broken up into two sections. I suggest consolidating sections 3.1.1 and 3.1.2 so that the equations that describe the method are in one place, and noting where appropriate what the formulation reduces to in the nominal phase case.
2. The authors claim that the MDP formulation is task-independent, but only demonstrate its application to a suite of door opening tasks. One thing I notice about this class of tasks is that progress in door opening tasks can be well-captured by a phase variable that can progress forward or backward (ex. if you fail to keep a spring loaded door open, it will shut, and look similar to the door at phase = 0). How would this formulation perform if task progress is less obviously captured by a phase variable, and failure can easily bring the robot/object states out of distribution of the initial demonstration (ex. if the task is stacking blocks, and at one point the stack topples and the blocks are now in a completely different arrangement)?
3. The work would be more impactful if demonstrated on real observations rather than ground-truth state.

**Quality Of The Limitations Section:**

3

**Questions For Rebuttal:**

1. Please provide an intuitive explanation of o1, o2, and o3 in the Observations section, as well as ensuring that all of the terms are defined (for example, the starred terms are not defined, q_0 and q_j are not defined, and the E frame is not defined).
2. In the actions section, please justify why sigma_1 is necessary.
3. I find the setup of Nominal Phase-Based Formulation in 3.1.1 and Adaptive Phase-Based Formulation in 3.1.2 to be unnecessarily confusing. Instead, I recommend presenting the full adaptive phase based-formulation, and simply highlighting (where appropriate) what the formulation would reduce to in the nominal phased-based case (like you do in lines 140-142 when describing the relationship between your adaptive phase rate and the nominal phase rate).
4. "The resulting dynamics induce a guiding effect during training similar to that achieved in reward-driven curriculum learning" (line 151-152), please provide some empirical or mathematical justification for this claim.

**Robotics Focus:**

4

**Summary Of Paper:**

Robust loco-manipulation achieved via sim2real RL, guided by a single demonstration. The key development is a state-dependent adaptive phase variable that captures task progress, which enables guidance from a demonstration while allowing adaptability in the presence of model mismatch and external disturbances. The authors demonstrate its applicability in the context of simulated and real door opening tasks.

**Summary Of Recommendation:**

I recommend this paper for acceptance because the adaptive phase formulation is interesting and effective for solving a suite of door opening tasks robustly, and because the questions of robustness and reduction of human effort in policy generation are topical in the robotics/machine learning research community right now. However, I suggest that the authors restructure the methods section to increase clarity and weaken their claims on the generality of the method (unless they demonstrate its applicability beyond door opening).

---

### Author Rebuttal · Authors · 2024-08-08

We thank the reviewers for their thorough evaluation and insightful comments on our submission. We appreciate the time and effort invested in providing feedback. While we address each of the reviewers' comments in detail in their respective responses, we summarize here the key changes made to the submission:

- Our proposed method is indeed dependent on the sufficiency of the demonstration to accomplish the task. Specifically, in the context of articulated object interactions where the demonstration is sufficient, our method effectively trains control policies that stabilize the execution around this demonstration while handling model mismatches and external disturbances. Additionally, it learns recovery behaviors from slippage and object handle misses-- scenarios not present in the demonstration itself. To clarify the scope of problems our method can solve, we have reworded the problem formulation accordingly.

- In response to the reviewers' suggestions for improving clarity in the methods section, we have rewritten Section 3 to focus primarily on our proposed formulation, mentioning the nominal formulation only when necessary. We hope these revisions address the concerns raised and enhance the clarity of our presentation.

- During the rebuttal phase, we conducted additional hardware experiments on the robot for all four loco-manipulation tasks considered in the paper (push and pull door traversing and dishwasher opening and closing). For each task, we performed continuous trials – where we executed the policy and resetted the scene manually before rerunning the policy (i.e., walking the robot to the front of the door at a random location, or resetting the dishwasher open/close state depending on the task). We have attached a separate video  (`hardware-deployment.mp4`) in the rebuttal zip file that shows these results and have added a discussion about them in Section 5.2. The video contains multiple runs over all four tasks, and for convenience, we provide the time-indexing for the tasks below:

  - 0:00 - Traversing a spring-loaded pull door
  - 1:54 - Traversing a spring-loaded push door
  - 3:33 - Dishwasher Opening
  - 4:17 - Dishwasher Closing

The rebuttal file includes the updated version of the manuscript along with the supplementary materials and videos. We are grateful for the constructive feedback and are confident that these enhancements will contribute to the value of our work. We look forward to the reviewers' further assessments.

---

### Decision · Program_Chairs · 2024-09-04

**Decision:**

Accept

**Comment:**

Update:
The authors have addressed reviewers' concerns during the rebuttal. There is an interesting idea to this paper that would benefit the CoRL community, and I found the new videos quite convincing.

====================================


Overall, the paper proposes an interesting method (adaptive phase variable) that can capture task progress for door-opening tasks, and its experimental results in simulation and real world are convincing.

However, the paper needs to tone down the argument about the generality of the proposed method -- as pointed out by several reviewers, it seems that there are several tasks where this approach is difficult to apply (see ones suggested by 9jLh). The paper needs to clearly articulate the class of problems that the method *can* solve, instead of claiming that the proposed method is task-agnostic. Lastly, I too agree with the reviewers that the method section is confusing to understand.